# From Pathogenesis to Therapeutics: A Review of 150 Years of Huntington’s Disease Research

**DOI:** 10.3390/ijms241613021

**Published:** 2023-08-21

**Authors:** Andrew Jiang, Renee R. Handley, Klaus Lehnert, Russell G. Snell

**Affiliations:** Applied Translational Genetics Group, Centre for Brain Research, School of Biological Sciences, The University of Auckland, Auckland 1010, New Zealand; r.handley@auckland.ac.nz (R.R.H.); klaus.lehnert@auckland.ac.nz (K.L.); r.snell@auckland.ac.nz (R.G.S.)

**Keywords:** Huntington’s disease, pathogenesis, therapeutics

## Abstract

Huntington’s disease (HD) is a debilitating neurodegenerative genetic disorder caused by an expanded polyglutamine-coding (CAG) trinucleotide repeat in the huntingtin (*HTT*) gene. HD behaves as a highly penetrant dominant disorder likely acting through a toxic gain of function by the mutant huntingtin protein. Widespread cellular degeneration of the medium spiny neurons of the caudate nucleus and putamen are responsible for the onset of symptomology that encompasses motor, cognitive, and behavioural abnormalities. Over the past 150 years of HD research since George Huntington published his description, a plethora of pathogenic mechanisms have been proposed with key themes including excitotoxicity, dopaminergic imbalance, mitochondrial dysfunction, metabolic defects, disruption of proteostasis, transcriptional dysregulation, and neuroinflammation. Despite the identification and characterisation of the causative gene and mutation and significant advances in our understanding of the cellular pathology in recent years, a disease-modifying intervention has not yet been clinically approved. This review includes an overview of Huntington’s disease, from its genetic aetiology to clinical presentation and its pathogenic manifestation. An updated view of molecular mechanisms and the latest therapeutic developments will also be discussed.

## 1. Introduction

Huntington’s disease (HD), previously known as Huntington’s chorea, was succinctly and accurately characterised in 1872 by physician George Huntington. He described the occurrence of a unique form of chorea characterised by “its hereditary nature… tendency to insanity… its coming on, at least as a grave disease, only in adult life” [1]. Subsequently, researchers have described a characteristic degeneration of striatal medium spiny neurons of the basal ganglia which is responsible for the manifestation of symptoms that encompasses motor, cognitive, and behavioural abnormalities [2]. There are now well-established clinical tests and, to a lesser extent, biomarkers that can ascertain both symptom onset and severity over the disease course [3,4]. The genetic aetiology was defined as an expansion of a polyglutamine coding repeat in exon 1 of the huntingtin gene [5], which lead to the development of an accurate genetic test for diagnosis [6]. Individuals with a polyglutamine repeat expansion > 39 CAGs display near-complete disease penetrance. However, the underlying molecular pathogenesis leading to the macroscopic neuronal loss remains to be comprehensively explained, and therapies in current clinical use were selected to alleviate symptoms rather than to delay or prevent the onset of the condition.

## 2. Clinical Implications

### 2.1. Prevalence and Diagnosis

HD occurs in all human populations worldwide; however, its prevalence varies between different ethnicities (Table 1). A relatively high prevalence of 10.6–13.7 per 100,000 affected is observed in individuals of Caucasian descent [7,8]. The prevalence of HD is drastically lower for Asian and African countries, with reports of 0.65, 0.42, and 0.01 per 100,000 in Japan [9], China [10,11], and South Africa [12], respectively. These differences in disease prevalence have largely been attributed to ethnic differences in the CAG repeat length. Populations with a higher prevalence of the disease exhibit longer CAG repeat lengths on average. An average CAG repeat length of 18.4–18.7 was found in people of European descent compared with 17.5–17.7 in East Asian populations [13]. The prevalence of HD has risen globally over the past two decades due to factors including longer lifespan [7,14], the emergence of an accurate genetic test [15], better care systems for HD individuals leading to longer disease survival times [16], and reduced stigma of diagnosis [17]. Another important factor is migration or travel where a mutation carrier (or carriers) produces offspring in an isolated community and the prevalence of the allele increases through drift of population expansion. Two notable examples of this are the Lake Maracaibo region of Venezuela, an extended kindred that was critical for the isolation of the HD gene [18], and Tasmania [19].

### 2.2. Clinical Presentation

HD symptomology can be categorised into a triad of progressive motor, cognitive, and psychiatric/behavioural disturbances. Motor dysfunction is characterised by the presence of involuntary movements (i.e., chorea, rigidity, dystonia) and the impairment of voluntary movements (i.e., difficulties in planning and completing tasks, dysarthria, dysphagia, and akinesia) [2,50]. Cognitive defects manifest as a decline in executive decision-making processes, particularly with goal-orientated behaviours. Impaired verbal learning and visuospatial complications are common and eventually widespread memory impairment is evident [2,50]. Neuropsychiatric deficits presented in HD include anxiety, irritability, depression, obsessive compulsive behaviour, aggression, apathy, and psychosis [2,51].

HD has traditionally been reported to affect only the central nervous system; however, emerging evidence suggests peripheral pathogenesis also occurs in organs including the heart, skeletal muscle, thyroid, liver, and digestive tract [52,53,54]. Weight loss is often noted in affected individuals, beginning presymptomatically and being maintained throughout the symptomatic stages of disease [55,56,57]. The onset of chorea and the energy expended through hyperkinetic activity does not explain the extent of the weight loss [50]. Other features include circadian rhythm abnormalities that result in a disrupted circadian rhythm, lower sleep efficiency, and delayed and shortened rapid eye movement (REM) sleep [58,59].

The average age at the onset of symptoms is between the ages of 35 and 40 years [2,4]. Disease duration is typically around 15–20 years after the onset of motor dysfunction, with the length of this period being independent of the CAG repeat length [60]. The clinical course is separated into 3 stages: first, the presymptomatic stage, with no detectable clinical abnormalities; second, a prodromal phase where individuals experience subtle changes in motor skills, cognitive ability, and behaviour, usually only noticeable by individuals in close contact with the affected individual; lastly, the manifest stage begins after a clinical diagnosis of HD is made [61]. Clinical diagnosis is currently evaluated through a combination of a confirmed family history or a positive genetic test and the onset of HD symptomatology. Motor impairment is defined by the Unified HD Rating Scale (UHDRS) total motor score (TMS) [62], which ranges from 0 (no motor deficits attributed to HD) to 4 (99% confidence that symptoms are HD-related). Cognitive tests and neuroimaging can aid the diagnosis, largely by excluding other similar disorders (HD phenocopies) [63].

### 2.3. Huntington’s Disease—A Disease of the Basal Ganglia

Neuropathologically, it is well established that the basal ganglia, particularly the medium spiny neurons of the striatum, are the most affected tissue in HD. The basal ganglia (basal nuclei) are a set of interconnected subcortical nuclei structures located deep within the centre of the cerebral hemispheres. It consists of the striatum (caudate nucleus and putamen), globus pallidus (interna (GPi) and externa (GPe)), subthalamic nucleus (STN), and substantia nigra (pars reticulata (SNr) and pars compacta (SNc)), which are interconnected with the cerebral cortex, thalamus, and brainstem (Figure 1). The basal ganglia are involved in a variety of functions including voluntary and involuntary movement, procedural learning, habit formation, cognition, and emotion [64,65]. Defects in the basal ganglia lead to many movement disorders such as parkinsonism and dyskinesias, as well as obsessive compulsive disorders and mood alterations [65].

## 3. Genetic Aetiology

HD is caused by a single mutation in the huntingtin (*HTT*) gene, located at chromosome 4p16.3, that encodes the protein huntingtin. Expansion of the CAG trinucleotide repeat in the polyglutamine-coding region of *HTT* exon 1 results in disease onset [5], with longer repeats associated with earlier onset [66,67]. The polyglutamine repeat expansion process behaves in a pure, dominant manner where both heterozygous and the rare homozygous HD patients show similar ages of onset. The addition of a second expanded polyglutamine *HTT* allele in the homozygotes yields no additional effect on onset age [68]. At the age of onset, disease progression does not significantly differ between homozygotes and heterozygotes [69,70]. HD falls under a wider group of polyglutamine repeat disorders that comprises dentatorubral-pallidoluysian atrophy (DRPLA), spinal and bulbar muscular atrophy/Kennedy’s disease (SBMA), and six forms of spinocerebellar ataxia (SCA 1, 2, 3 (Machado-Joseph disease), 6, 7 and 17) [71,72].

The exon 1 polyglutamine-coding repeat is polymorphic in length, and length distribution varies between ethnic groups. Repeat lengths of 26 or less are considered within the normal range. Repeats of 27–35 are deemed intermediate and on transmission may expand into the disease-causing range. Inheritance of a repeat length of 36–39 may result in the development of HD, but not in all cases (reduced penetrance alleles). The inheritance of repeat lengths of 40 CAG units or longer are almost completely penetrant and lead to the onset of HD symptomatology [4,50]. Germline instability in the polyglutamine repeat, known as anticipation, can occur, resulting in an earlier age of onset in offspring, normally on paternal transmission [73,74]. A rare, early-onset form of HD known as juvenile HD may present in individuals below the age of 20. Juvenile HD is generally associated with a polyglutamine repeat length of >60 CAGs, with some childhood cases presenting with ~80 CAGs [18]. The prevalence of juvenile HD is estimated at 5% of all HD cases [75]. Individuals display many phenotypic similarities with adult-onset disease; however, chorea is often rare, and the predominate phenotype is rigidity, akinesia, and bradykinesia [75,76]. Progressive cognitive deficits are also common in juvenile HD, with mental retardation and learning disabilities in childhood cases [77].

Somatic repeat instability has also been described, predominantly in the HD brain, with the brain striatum and cortical regions being the most affected [78,79,80,81,82]. The expanded CAG tract likely results in the formation of unusual DNA structures, which induces an increased propensity for DNA mismatch repair errors during replication, DNA damage, and transcription [83,84]. Several DNA mismatch repair genes including *FAN1*, *MSH1*, and *MSH3* have been shown to induce somatic instability including large expansions of the CAG tract in mouse models of HD [85,86,87] and cell lines derived from HD patients [88,89]. Interest in somatic repeat instability increased with the results of the genetic modifier studies discussed in the next section.

### 3.1. Genetic Disease Modifiers

The polyglutamine repeat length accounts for approximately 60% of the total variation in the age of onset, with the remaining variance attributed to other genetic modifiers and environmental or stochastic effects [90]. A genome wide search for genetic modifiers of the age at onset was performed by the Genetic Modifiers of Huntington’s Disease (GeM-HD) Consortium using a dataset comprising cases from multiple HD cohorts including HSG PHAROS [91], COHORT [92], TREND-HD [93], PREDICT-HD [94], REGISTRY [95], ENROLL-HD [96], and linkage studies [97]. Several modifier loci were identified that associated with the residual age of onset (not explained by the HD polyglutamine repeat length) and that do not have detectable effects in the absence of the polyglutamine expansion [98]. These included effects at chromosome 2 (*PMS1*), chromosome 3 (*MLH1*), chromosome 5 (*MSH3*/*DHFR*, *TCERG1*), chromosome 7 (*PMS2*), chromosome 8 (*RRM2B*, *UBR5*), chromosome 11 (*CCDC82*), chromosome 15 (*FAN1*), and chromosome 19 (*LIG1*). Many of the candidate genes underlying these significant loci are involved in DNA repair (*FAN1*, *LIG1*, *MLH1*, *MSH3*, *PMS1*, and *PMS2*) and contribute to somatic instability of the HD CAG repeat [98,99,100]. This work has revealed new potential drug targets for the treatment of HD [101].

### 3.2. Huntingtin Structure and Function

The huntingtin gene (*HTT*) contains 67 exons and is processed to form two alternate mRNA transcripts of 10,366 bp and 13,711 bp that differ by the exclusion/inclusion of a proportion of the 3′UTR (Figure 2) [102]. In addition, alternative splicing generates several huntingtin isoforms all with relatively low abundance, with transcripts lacking exons 10, 12, 29, and 46 or retaining a 57 bp portion of intron 28. Interestingly, expression of these rare variants is upregulated in HD and may impact gene functionality [103,104]. The protein encoded by *HTT* (wild-type polyglutamine repeat length of 23 CAG) consists of 3144 amino acids with a molecular weight of 348 kD [105]. *HTT* is ubiquitously expressed throughout most human tissues, but is highest in the nervous system, liver, and testes (http://www.gtexportal.org/home/gene/HTT (accessed on 14 January 2023), http://www.proteinatlas.org/ENSG00000197386-HTT/tissue (accessed on 14 January 2023).

The huntingtin protein is generally well conserved throughout mammalian species, suggesting it plays a crucial role in species viability and functionality. The protein is comprised of an N-terminal region that includes: (1) a 17 amino acid amphipathic alpha helical structure known as HTT^NT^, which is important for HTT nuclear export and targeting to the endoplasmic reticulum; (2) the polyglutamine repeat (polyQ) responsible for disease onset; and (3) a polymorphic proline rich domain (PRD) that facilitates protein–protein interactions with proteins that contain tryptophans (WW) or src homology 3 domains. The proline rich domain forms a rigid proline–proline helical structure, which may play a role in mutant huntingtin protein aggregation [105]. Proteolytic cleavage of N-terminal fragments (HTT^NT^ + polyQ + PRD) by endoproteases, caspases, and calpains occurs more frequently in HD, and it is hypothesised that these N-terminal fragments are involved in disease pathogenesis [106,107]. The remainder of the protein is organised into clusters of anti-parallel alpha-helical HEAT (huntingtin elongation factor 3, protein phosphatase 2A, and yeast kinase TOR1) repeats, which function as scaffold motifs for other protein macromolecules [108].

Huntingtin protein has been shown to interact with more than 350 confirmed binding partners (http://cbdm-01.zdv.uni-mainz.de/~mschaefer/hippie/query.php?s=HTT, accessed on 14 January 2023) [109] and is involved in multiple cellular processes that include vesicle trafficking, endocytosis, cell adhesion, metabolism, transcription, RNA processing, and protein turnover [105,110]. Post-translational modification sites dispersed throughout huntingtin regulate its activity and interaction with other macromolecules. These include sites for proteolysis, phosphorylation, palmitoylation, ubiquitylation, sumoylation, and acetylation. Given the variety of post-translational modifications, numerous binding partners, and intramolecular interactions, the huntingtin protein is able to adopt >100 structurally distinguishable 3D conformations, many of which have been proposed to confer toxicity [110,111].

## 4. Hallmarks of Disease

### 4.1. Neurodegeneration of Striatal GABAergic Medium Spiny Neurons

The death of GABAergic medium spiny neurons (MSN) located in the striatum and projecting into the globus pallidus externa is a hallmark of HD. GABAergic MSNs comprise approximately 95% of the neurons in the striatum, and over the course of HD, >90% of these neurons degenerate. In contrast, interneurons, which form the other 5% of the neurons of the striatum are relatively spared [112,113]. The preferential susceptibility of MSN degeneration is debated, although prevailing hypotheses include the observation of increased somatic instability in the striatum compared with other brain regions [78,79,80,81,82], preferential glutamate excitotoxicity of MSN, vulnerability for mitochondrial dysfunction due to an increased metabolic demand in MSNs, and altered striatal brain-derived neurotrophic factor (BDNF) activity (these themes will be further discussed in the molecular mechanisms and therapeutic development section of this review) [114,115,116,117,118,119].

MSNs are dispersed throughout both the direct and indirect pathways, and their degeneration greatly affects the symptoms presented (Figure 2). The initial loss of MSNs of the indirect pathway results in a hyperkinetic state and the onset of involuntary movements and chorea. This is followed by the loss of MSNs of the direct pathway later on in the disease course, resulting in a hypokinetic phenotype and the onset of motor impersistance and akinesia [113]. To characterise the extent of striatal neuropathy and the loss of MSNs, a post-mortem histopathological grading system was developed by Vonsattel and colleagues [120]. The system starts from grade 0, with no evidence of macroscopic or microscopic cell loss, and proceeds to grade 3/4, where a marked decrease in the size of the striatum and globus pallidus with extensive widespread neuronal loss is observed. A grade 4 classification indicates >90% neuronal loss in the striatum [120,121,122].

### 4.2. Neurodegeneration in Other Brain Regions

While HD is largely characterised by the death of MSN in the striatum, extensive neurodegeneration is also seen in multiple other brain regions. Progressive atrophy occurs throughout the cerebral cortex, with cortical pyramidal neurons in cortical layers III, V, and VI being the most affected [123]. This has been further confirmed by in vivo high-resolution magnetic resonance imaging and automated surface reconstruction [124,125]. Interestingly, the pattern of cortical thinning was associated with the patient’s symptom profile. Neurodegeneration of primarily pyramidal neurons of the motor cortex results in onset of predominant HD motor symptoms. In contrast, neurodegeneration of cortical pyramidal neurons of the cingulate gyrus region, which processes emotions, is mainly associated with mood alterations [126,127,128,129]. Other affected areas include the cerebellum [130], globus pallidus [131,132], amygdala [133,134,135,136], and hippocampus [137]. The evidence to support neurodegeneration in these areas has been disputed [138,139,140,141].

### 4.3. Toxic or Neuroprotective mHTT Inclusions?

The propensity for mutant huntingtin (mHTT) protein to aggregate and form inclusions is another core hallmark of HD. The 17-residue N-terminal sequence of *HTT*, termed HTT^NT^ promotes protein aggregation through the formation of prefibrillar oligomers of antiparallel polyglutamine beta sheet structures, forming an amyloid-like structure similar to those found in Alzheimer’s disease and prion diseases [142,143]. The majority of protein aggregates are comprised primarily of N-terminal fragments generated through an aberrant cleavage of expanded polyQ *mHTT* exon 1 [105,144,145]. These N-terminal fragments have also been shown to translocate both extracellularly [146,147] and within nuclei [148,149].

Whether the aggregation event itself is pathogenic in HD remains ambiguous. A popular hypothesis is that the inclusions are cytotoxic and that accumulation above a certain threshold triggers pathogenic events caused by either the aggregate itself or through the depletion of wild-type huntingtin protein within the cytosol [4,105]. The underlying mechanism to achieve this is unknown, possibly due to the complex nature of mutant huntingtin toxicity, which is dependent on the disease stage, length of polyglutamine repeat, cellular location, and conformational state. Alternatively, instead of a threshold requirement for toxicity, cytotoxicity may be induced by specific small mutant huntingtin oligomeric species that have adopted specific conformational states [150,151]. Conversely, aggregation may not have such a pernicious effect. Indeed, researchers have observed a higher frequency of inclusion bodies in the cerebral cortex, which exhibits a lower degree of cell loss compared to striatal regions. This may potentially be attributed to the inability to detect aggregates in already degenerated neurons [152,153]; however, in vivo studies of cultured striatal neurons indicated that the presence of intranuclear inclusions did not correlate with cell death [106]. Additionally, a transgenic HD mouse model showcased the development of HD-related motor and behavioural abnormalities that preceded the formation of inclusions [154]. Other studies have suggested that the inclusion bodies may be neuroprotective by reducing the bioavailability of toxic soluble mutant huntingtin species [155]. Investigations in HD mouse models further supports this theory with several transgenic [156,157] and knock-in HD mouse models [158,159,160] displaying widespread inclusions in the absence of overt motor and behavioural deficits or neurodegeneration.

An emerging area of research is phenomenon of repeat-associated non-AUG (RAN) translation, where translation bypasses the requirement for an AUG start codon and results in protein translation from all three different reading frames [161]. The efficiency of RAN translation is dependent on the repeat length [162]. RAN-translated products have been detected in HD patients and include polyalanine, polyserine, polyleucine, and polycystine proteins. These proteins were observed to accumulate and aggregate in various brain regions in a CAG length-dependent manner [163]. Experiments in neuronal and glial cell models showed that the presence of RAN proteins was observed to reduce cell survival [163]. However, whether RAN translation induced by the *HTT* CAG repeat plays a crucial role in pathogenesis remains to be seen [164].

### 4.4. HD Biomarkers

A biological marker (biomarker) that can reliably track disease progression and evaluate pharmacological responses to drugs is desirable not only for prognostic purposes but also because it can provide a surrogate end point for clinical studies. As genetic testing is available for HD, biomarker changes would therefore ideally be detectable in pre-symptomatic individuals, reflecting the disease process prior to functional decline [3,4]. Longitudinal observational studies including TRACK-HD [165] and PREDICT-HD [94] have been assembled with the aim of identifying reliable biomarkers.

A multitude of HD biomarkers have been suggested that span biochemical, genetic, imaging, and clinical disciplines [3,166]. The measurement of neurofilament light chain (NfL) and mutant huntingtin protein levels in cerebrospinal fluid (CSF) have been the most promising early biochemical biomarkers of HD to date. Increased levels of NfL have been identified in CSF and plasma from individuals with HD compared with control subjects [167,168,169,170], and this correlates with clinical measures of dysfunction and brain atrophy [168,169,170]. Levels of mutant huntingtin protein are also reported to be elevated in the CSF of HD individuals compared with controls, with the ability to distinguish between premanifest and manifest HD cohorts [146,171]. Other biochemical biomarkers studied include neuropeptide Y and tau measured in cerebrospinal fluid and brain-derived neurotrophic factor (BDNF), cholesterol, glutathione, Hb and α1-microglobulin, lactate, lipid peroxide, mutant huntingtin protein, miRNA, neurofilament heavy chain, 8OHdG, SERCA2, VEGF, TGF-β, and uric acid in serum [172]. Imaging biomarker investigations include evaluations of the degree of brain atrophy using volumetric magnetic resonance imaging (MRI) and other MRI modalities (functional MRI, magnetic resonance spectroscopy, transcranial magnetic stimulation, and brain iron) [166]. Positron emission tomography (PET) scans have also been utilised to infer brain glucose metabolic activity, PDE10A activity, and striatal dopamine D1 and D2 receptor activity [173]. Clinical biomarker assessments include the ascertainment of motor function through grip force, hand tapping, oculomotor change, and UHDRS total motor score. Cognitive measures include language, mood, and neuropsychological assessments. Other measures including auditory time perception and electrophysiological testing have also been utilised [4].

### 4.5. HD as a Peripheral Disease?

Alongside the pronounced HD lesion in the striatum, evidence to support a role for peripheral metabolic pathways in pathogenesis is becoming increasingly apparent. As discussed, HD patients experience weight loss despite increased caloric intake [56,174]. In conjunction with skeletal muscle atrophy, glucose intolerance, and gastrointestinal abnormalities, a widespread metabolic disruption has been proposed [53]. Metabolite analysis of HD human blood samples show altered levels of aliphatic amino acids and fatty acid breakdown [175]. Another emerging area of interest is the observation of elevated levels of urea in post-mortem human brains [176]. Similar observations were also seen for HD mouse models [177] and an ovine transgenic HD model expressing 73 polyglutamine repeats [178]. Analysis of liver RNA-seq data from both the HD mouse and sheep models indicates a perturbation of urea cycle genes [177,179]. The link between liver dysfunction and neurological deficits is not novel, as evidenced by individuals with uremic encephalopathy [180] and urea cycle disorders [181]. Urea cycle disorders result in a build-up of ammonia which is implicated in neurotoxicity, and individuals often display symptomology with phenotypic resemblance to HD [181,182].

## 5. Animal Models

HD mouse models have been extensively used and include early toxin-induced models (Quinolinic acid, 3-Nitropropionic), N-terminal *HTT* models (R6/2, N171-82Q/N118-82Q/N586-82Q, HD94, shortstop, HD150QG/HD190QG), and more recent full-length *HTT* (BACHD, YAC72/128, Hu97/18) and knock-in models (HdhQ50/92/111/140/150, zQ175) (overview of HD models in Table 2). A comprehensive review of animal models of HD is outside the scope of this review, however, there are many excellent reviews on this subject [183,184,185].

Nevertheless, there are several unavoidable limitations with the use of small animal models. HD is a late-onset disorder with an approximate 40-year presymptomatic period in humans. The short lifespan of mice restricts their potential for extended assessment and ability to mimic disease progression. Moreover, biological differences in the brain including size, lack of a complex gyrencephalic structure, and complete absence of some brain regions limit the application of mice modelling human neurodegenerative disease. Therefore, recent attention has focused on developing large mammalian HD models including sheep [186], pigs [187,188,189,190], and non-human primates [191,192,193] that are more anatomically similar to humans and have longer lifespans which may capture more complex features of the disease. The ideal animal model would need to not only replicate the neuropathology and symptomology of the disease but also showcase comparable disease progression.

**Table 2 ijms-24-13021-t002:** Huntington’s disease animal models.

Toxin Models								
Toxic Compound	Species	Type of Administration	Mechanism of Action	HD-Related Phenotypes				Reference
				Motor	Neuronal Loss	mHTT Aggreg.	Early Death ^1^	
Quinolinic acid	*Rattus norvegicus*, *Mus musculus*, non-human primates	Intrastriatal injections	Glutamate excitotoxicity	✓	✓	X	✓	[194,195,196,197,198,199,200,201,202,203,204]
3-Nitropropionic	*Rattus norvegicus*, *Mus musculus*, non-human primates	Systemic, intrastriatal injections	Mitochondrial dysfunction via succinate dehydrogenase inhibition	✓	✓	X	✓	[118,119,195,205,206,207,208,209,210,211,212,213]
**Mouse Models**								
**Animal Model**	**Species**	**Construct + Promoter**	**Polyglutamine Repeat Size**	**HD-Related Phenotypes**				**Reference**
				**Motor**	**Neuronal Loss**	**mHTT Aggreg.**	**Early Death ^1^**	
(a) Random integration transgenic models								
R6/2	*Mus musculus*	1 Kb 5′UTR + exon 1 of human *HTT* + 262 bp of intron 1 Human *HTT* promoter	144	✓	✓	✓	✓	[214,215,216,217,218]
R6/1	*Mus musculus*	1 Kb 5′UTR + exon 1 of human *HTT* + 262 bp of intron 1 Human *HTT* promoter	116	✓	✓	✓	✓	[216]
N171-82Q	*Mus musculus*	First 171 amino acids of human *HTT* cDNA Mouse *Prp* promoter	82	✓	✓	✓	✓	[219,220]
HD94	*Mus musculus*	100 bp UTR + human/mouse exon 1 (chimeric) + 600 bp intron 1. *CamKIIa* promoter	94	✓	✓	✓	✓	[221]
Shortstop	*Mus musculus*	First 171 amino acids of human *HTT* Human *HTT* promoter	128	X	X	✓	X	[156]
N118-82Q	*Mus musculus*	First 118 amino acids of human *HTT* cDNA Mouse *Prp* promoter	82	X	X	✓	✓	[222]
N586-82Q	*Mus musculus*	First 586 amino acids of human *HTT* cDNA Mouse *Prp* promoter	82	✓	✓	✓	✓	[222]
HTT-160Q-31	*Mus musculus*	First 208 amino acids of human *HTT* cDNA *GFAP* promoter	160	✓	X	✓	✓	[117]
Ubi-G-HTT84Q	*Mus musculus*	First 153 amino acids of human *HTT* cDNA fused to EGFP Ubiquitin promoter	84	✓	X	✓	✓	[223]
HD150QG, HD190QG	*Mus musculus*	First 67 amino acids of human *HTT* cDNA fused to EGFP Human *HTT* promoter	150, 190	✓	X	✓	✓	[224]
BACHD	*Mus musculus*	Full length human *HTT* Human *HTT* promoter	97	✓	✓	✓	X	[225]
YAC72	*Mus musculus*	Full length human *HTT* Human *HTT* promoter	72	X	✓	✓	X	[157]
YAC128	*Mus musculus*	Full length human *HTT* Human *HTT* promoter	128	✓	✓	✓	X	[226,227]
Hu97/18	*Mus musculus*	Full length human *HTT*, two copies of the full length human *HTT* alleles Human *HTT* promoter	97	✓	✓	X	X	[228]
iFL148Q	*Mus musculus*	Inducible full length human *HTT* cDNA Mouse *Prp* promoter	148	✓	X	✓	✓	[229]
BACHD rats	*Rattus norvegicus*	Full length human *HTT* Human *HTT* promoter	97	✓	✓	✓	X	[230,231]
HD51	*Rattus norvegicus*	First 727 amino acids of rat *HTT* cDNA Rat *HTT* promoter	51	✓	✓	✓	X	[232,233,234]
(b) Knock-in models								
HdhQ50	*Mus musculus*	Replacement of mouse *Hdh* exon 1 with human *HTT* with 50 CAG repeats Mouse *HTT* promoter	50	X	X	X	X	[160]
HdhQ92, HdhQ111, HdhQ140, HdhQ150	*Mus musculus*	Replacement of mouse *Hdh* exon 1 with human *HTT* with 92, 111, 140, 150 CAG repeats. Mouse *HTT* promoter	92, 111, 140, 150	✓	✓	✓	X	[154,158,159,235,236,237]
zQ175	*Mus musculus*	Replacement of mouse *Hdh* exon 1 with human *HTT* with 188 CAG repeats Mouse *HTT* promoter	188	✓	✓	✓	✓	[238]
(c) Knock-out models								
**Animal Model**		**Species**	**HD-Related Phenotypes**					**Reference**
Hdhex5 (mutation at mouse *HTT* exon 5)		*Mus musculus*	-Homozygous knockouts are embryonically lethal at day 8.5 -Heterozygous animals show motor defects and neuronal loss					[239]
Deletion of exon 4 and 5 in mouse HTT		*Mus musculus*	-Homozygous knockouts are embryonically lethal at day 8.5 -Heterozygous animals showcase no overt phenotype and are phenotypically normal					[240]
Deletion of a 2.8 kb of 5′ flanking sequence, entire exon 1 and 0.6 kb of 3′ intron of mouse HTT		*Mus musculus*	-Homozygous knockouts are embryonically lethal at day 8.5–10.5 -Heterozygous animals showcase no overt phenotype and are phenotypically normal					[241]
**Large Mammalian Models**								
**Animal Model**	**Species**	**Construct + Promoter**	**Polyglutamine Repeat Size**	**HD-Related Phenotypes**				**Reference**
				**Motor**	**Neuronal Loss**	**mHTT Aggreg.**	**Early Death ^1^**	
Transgenic sheep (*OVT73*)	*Ovis aries*	Full length human *HTT* cDNA + 1.1 kb human *HTT* genomic DNA upstream of exon 1 + 3′ bovine growth hormone UTR Human *HTT* promoter	73	X	X	✓	X	[59,178,186,242,243]
Transgenic pig #1	*Sus domesticus*	First 1100 amino acids of *Sus domesticus HTT* cDNA Rat neuron-specific enolase (NSE) promoter	75	No reports of these animals				[187]
Transgenic pig #2	*Sus domesticus*	First 208 amino acids of human *HTT* cDNA Cytomegalovirus enhancer/chicken beta-actin promoter	105	✓	✓	✓	✓	[188]
Transgenic pig #3	*Sus domesticus*	First 548 amino acids of human *HTT* cDNA Human *HTT* promoter	124	✓	✓	✓	X	[189,244]
Knock-in pig	*Sus domesticus*	Replacement of pig *HTT* exon 1 with human *HTT* exon 1 with 150 CAG repeats Human *HTT* promoter	150	✓	✓	✓	✓	[190]
Transgenic monkeys #1	*Macaca fascicularis*	Lentiviral injection of the first 171 amino acids of human *HTT* cDNA into rhesus macaque’s striatum PGK-1 lentiviral promoter	82	✓	X	✓	X	[191]
Transgenic monkeys #2	*Macaca mulatta*	Human *HTT* exon 1 + GFP gene Human Ubiquitin promoter	84	✓	✓	✓	✓	[192,245]
Transgenic monkeys #3 (rHD6, rHD7, rHD8)	*Macaca mulatta*	First 508 amino acids of human *HTT* cDNA Human *HTT* promoter	67, 70, 72	✓	✓	✓	X	[193]

✓ Observed in study; X Not observed in study. ^1^ Occurrence of early death within the timeframe of the study.

## 6. Molecular Mechanisms and Therapeutic Development

An assortment of mechanisms for HD pathogenesis have been proposed and investigated. Two core hypotheses constitute the rationale for research and therapeutic design: a gain of aberrant toxicity from *mHTT* expression and a dominant loss of normal activity from wild-type *HTT* as a result of *mHTT* expression. An overview of the current proposed pathogenic mechanisms and drug development based on these mechanistic pathways will be discussed (summarised in Figure 3 and Table 3).

### 6.1. Excitotoxicity-Induced Neurodegeneration

Glutamate excitotoxicity as an initiator of neuronal death has been a longstanding hypothesis and research focus in neurodegenerative disorders [246,247,248,249]. In HD, it has been proposed that the neurodegeneration of the medium spiny neurons of the striatum is caused by excess glutamate in the synapses resulting from a lack of synaptic glutamate clearance and an increased glutamate release. An observed reduction in expression of astrocytic glutamate uptake receptors, GLT1, and GLAST has been reported in the striatum of HD mouse models and human post-mortem tissue, which may explain the elevated glutamate levels [116,117]. Excess synaptic glutamate has been proposed to lead to increased signalling of post-synaptic N-methyl-d-aspartate (NMDA) receptors, α-amino-3-hydroxy-5-methyl-4-isoxazolepropionic acid (AMPA) receptors, and kainate receptors. Consequently, this may lead to an influx of calcium ions, chronic membrane depolarisation, oxidative stress, and activation of cell death pathways [246,247,248,249,250]. Evidence for NMDA receptor-mediated excitotoxicity was first shown through striatal injection of glutamate receptor agonists (glutamic acid, kainic acid, quinolinic acid) in rodents and non-human primates, resulting in medium spiny neuronal degeneration and HD motor dysfunction [196,197,198,200,201,251,252]. Further, studies investigating several HD mouse models provided evidence for an association between excessive NMDA receptor expression, receptor currents, and striatal degeneration [253,254,255]. Ionotropic glutamate receptors were also most pronounced in striatal medium spiny neurons compared to striatal interneurons, which may explain the preferential neurodegeneration of medium spiny neurons [256].

#### Treatments Targeting Excitotoxicity

NMDA receptor antagonists have been tested in clinical trials for their ability to reduce glutamate receptor overactivation. These compounds have included Memantine, an NMDA receptor antagonist that is FDA approved for the treatment of dementia symptoms in Alzheimer’s disease [257]. Memantine first showed efficacy in the 3-nitropropionic mouse model, where intraperitoneal injections of Memantine resulted in reduced striatal degeneration and improved motor and cognitive function [258]. A phase IV clinical trial was conducted involving 50 symptomatic HD patients that received either memantine or a placebo and were followed up with over 3 months. Unfortunately, the study showed no improvement to motor function between the treatment and control groups, although the researchers noted that longer studies are likely required to validate these findings [259]. Another NMDA receptor antagonist, Dimebon (laterepirdine), failed to showcase improvement in motor and cognitive function in a placebo-controlled phase III clinical trial involving 350 HD patients over the course of 6 months [260]. The NMDA receptor antagonist, amantadine was trialled in HD individuals for its efficacy in reducing parkinsonian dyskinesias [261,262]. The efficacy of amantadine in clinical trials has been mixed, with some trials reporting improvement to motor function and others indicating no significant differences [263,264,265].

Other treatments targeting excitotoxicity have been developed to limit the availability of extracellular synaptic glutamate. Excessive glutamate release may be diminished by blocking voltage-gated sodium channels. The sodium channel blocker Riluzole was investigated in HD patients for its beneficial effects in mediating excitotoxicity in mouse models [246]. However, a phase III placebo-controlled clinical trial conducted with 537 HD patients over a three-year period concluded that riluzole had no benefit for the treatment of HD, with no improvements to motor, cognitive, or behavioural function observed [266]. Another sodium channel blocker, BN82451, was shown to improve motor function, decrease atrophy, and extended survival in R6/2 HD mice [267]. However, a phase II placebo-controlled clinical trial involving 17 patients was terminated due to subject recruitment problems (www.clinicaltrials.gov, identifier NCT02231580, accessed on 17 August 2023).

Overall, NMDA antagonists and glutamate-lowering drugs have showed limited, sustained efficacy in the improvement of HD motor and cognitive deficiencies. However, several of these studies were underpowered, with a limited number of participants treated over a short period of time. The trials also recruited mainly symptomatic HD patients, for whom widespread degeneration of neurons would have already occurred. Assessment of the efficacy of treatments targeting excitotoxicity may be confounded by the consequences of widespread catastrophe brought upon by the cell death cascade reaching a point where recovery becomes unattainable. With evidence that suggests excitotoxicity may begin early in the disease phase [268,269,270], future clinical trials should consider whether administration of these therapeutics at an earlier disease stage may result in improved efficacy compared to what is currently observed.

### 6.2. Dopaminergic Imbalance

The dopaminergic pathway is thought to be dysfunctional in HD and responsible for movement-related symptoms including chorea, rigidity, and akinesia [271]. Studies examining the cerebrospinal fluid of HD patients have indicated an increase in dopamine levels during early stages of the disease, correlating to the onset of chorea [272]. Interestingly, dopamine levels decrease throughout disease course, and post-mortem HD brains show reduced levels of caudate and striatal dopamine correlating with rigidity and akinesia symptoms present during late stages of the disease [273]. PET studies have shown a significant decrease in dopamine availability in akinetic patients, resulting in reduced binding to striatal dopamine D1 and D2 receptors [274,275], dopamine transporter (DAT) [276], and vesicular monoamine transporter 2 (VMAT2) [277]. Findings in HD mouse models agree with human studies showing an initial increase in dopamine levels and dopamine release and binding to D1 and D2 receptors, followed by a progressive reduction with disease progression [278,279,280,281,282]. The exact mechanisms of dopamine regulation in the HD basal ganglia are unclear and a question remains as to whether dopamine alterations are a cause or consequence of the widespread neurodegeneration.

#### Treatments Targeting the Dopaminergic Pathway

Two FDA-approved drugs for the symptomatic treatment of HD-mediated chorea, tetrabenazine and deutetrabenazine, target the dopaminergic pathway through inhibition of the vesicular monoamine transporter (VMAT2 inhibitor), resulting in decreased bioavailability of dopamine in synapses and reduced dopamine signalling. Both tetrabenazine and deutetrabenazine ameliorated chorea and other motor symptoms in an HD mouse model [283]. These results were similarly replicated in human clinical trials showcasing improved motor function [284,285,286]. The deuterated form of tetrabenazine, deutetrabenazine improved on the half-life of tetrabenazine and allowed for administration at lower doses and reduced adverse effects [286,287]. Recently, another VMAT2 inhibitor, valbenazine, has shown success in a phase III clinical trial with preliminary results indicating a significant improvement in motor function (www.clinicaltrials.gov, identifier NCT04102579, accessed on 17 August 2023) [288].

Antipsychotic drugs are commonly prescribed off-label for HD chorea, psychiatric problems, sleep dysfunction, and weight loss despite no consistent evidence to support their therapeutic use [289]. Risperidone, generally used in the treatment of schizophrenia and bipolar disorders [290], is currently being assessed in a phase II clinical trial involving 12 participants (www.clinicaltrials.gov, identifier NCT04201834, accessed on 17 August 2023). A completed phase III clinical trial comparing tetrabenazine with two antipsychotics, olanzapine and tiapride, is expected to publish results (www.clinicaltrials.gov, identifier NCT00632645, accessed on 17 August 2023). Another antipsychotic, rolipram, was shown to be neuroprotective in an HD mouse model, and was hypothesised to act through its inhibition of phosphodiesterase and the second messenger dopamine cascade [291].

### 6.3. Mitochondrial Dysfunction

Evidence for mitochondrial dysfunction in HD stems from the observation that systemic administration of 3-nitropropionic acid, an irreversible inhibitor of complex II of the mitochondrial electron transport chain in both rodents and non-human primates, reproduced many of the behavioural and anatomical characteristics of HD [118,119]. Mutant huntingtin protein also appears to directly sequester the intracellular transport machinery and potentially interfere with both anterograde and retrograde trafficking of mitochondria to sites of high energy demand such as neuronal dendrites, axons, and synapses, culminating in overall synaptic degeneration and neurodegeneration [292,293,294]. As previously mentioned, metabolic energy deficits have been reported in HD patients, often shown as weight loss. Increased lactic acid and decreased glucose metabolism were observed in HD patients [295,296,297], and a reduction in circulating levels of branched chain amino acids typically involved in energy homeostasis was seen that is correlated with weight loss [174]. These changes were seen early in the disease before the onset of symptoms and suggest that metabolic deficits in HD may precede neuropathology and clinical symptoms [174].

In HD patient-derived cell lines and HD mouse models, a number of studies showcase abnormal ATP:ADP and phosphocreatine:inorganic phosphate (PCr:Pi) ratios, likely due to an impaired mitochondrial electron flow [298,299,300]. Isolated mitochondria from HD mouse models and lymphoblasts from HD patients showcased a depolarised mitochondrial membrane that became progressively disrupted with increasing polyglutamine repeat length [301,302]. A decrease in the activity of electron transport chain complex II (succinate dehydrogenase) and complex IV (cytochrome c oxidase) is suggested to explain the membrane depolarisation [303]. Further, it has been proposed that an altered electron flow through the compromised mitochondrial complexes may promote the formation of reactive oxygen species (ROS) including superoxide (O_2_•−), hydrogen peroxide (H_2_O_2_), hydroxyl radical (•OH), and peroxynitrite (ONOO−) [304], which leads to activation of downstream apoptotic mechanisms of cell death.

#### Treatments Targeting Mitochondrial Dysfunction

Assessment of therapeutics targeting HD mitochondrial dysfunction have been confounded by difficulties in optimal dosing, trial length, and accurate measurement of target engagement. Eicosapentaenoic acid, an omega-3 fatty acid used to treat hypertriglyceridemia, showcased improvements to motor and behavioural deficits in an HD mouse model [305]. However, a phase III placebo-controlled clinical trial involving 300 patients showed no improvement to UHDRS motor scores over 6 months with no alterations in neurodegeneration [93,306]. Similarly, coenzyme Q10 supplementation was found to be neuroprotective in an HD mouse model displaying improvements in motor deficits, delayed striatal atrophy, and prolonged lifespans [307]. However, two independent large placebo-controlled clinical trials showed no significant slowing of disease progression at all dosages studied [308], with one of the studies (2CARE) terminated early due to the occurrence of severe adverse effects [309]. Creatine, which contains antioxidant properties, delayed striatal degeneration and reduced mutant huntingtin protein aggregation in an HD mouse model [310]. Again, clinical trials were unsatisfactory and a phase III placebo-controlled clinical trial involving 553 patients followed up to 48 months showed no influence on motor and cognitive deficits [311,312]. The cytoprotective peptide SBT-20 acts to prevent lipid peroxidation and initiation of apoptosis. SBT-20 was effective in protecting against mutant huntingtin-induced mitochondrial and synaptic damage in cultured HD neurons [313]. However, no significant differences in mitochondrial function were seen between treatment and placebo groups in phase I/II clinical trials [314].

Some promising results have been reported for Triheptanoin, a triglyceride that catabolises into substrates for the Krebs cycle, targeting the amelioration of HD metabolic deficits. Following one month of treatment, the ratio between inorganic phosphate and phosphocreatine levels that is disrupted in HD [298,299,300] was remedied, along with improvement in motor deficits [315]. A phase II placebo-controlled study involving 100 participants followed over 12 months was recently completed, evaluating triheptanoin in pre-manifest HD individuals (www.clinicaltrials.gov, identifier NCT02453061, accessed on 17 August 2023). Resveratrol, an antifungal agent proposed to inhibit p53-mediated mitochondrial apoptotic oxidative stress is currently being assessed in a phase III placebo-controlled clinical trial involving 102 patients followed over 12 months (www.clinicaltrials.gov, identifier NCT02336633, accessed on 17 August 2023) [316]. The PPARα agonist Fenofibrate is also currently being evaluated in a phase II placebo-controlled clinical trial involving 20 participants followed over a period of 6 months for its ability to induce PGC-1α expression levels involved in mitochondrial fission and fusion (www.clinicaltrials.gov, identifier NCT03515213, accessed on 17 August 2023).

### 6.4. Neuroinflammation

Neuroinflammation in the scope of neurodegenerative disorders describes the activation of glial cells, notably astrocytes and microglia, which respond to neuronal damage and act to remove the dead cells and other constituents by phagocytosis. However, this beneficial role may be limited, as chronic neuroinflammation, as in the case of HD, results in the accumulation of soluble pro-inflammatory mediators including cytokines, prostaglandins, and nitric oxide that further exacerbate cellular degeneration [317]. Evidence for neuroinflammation in HD has been presented as the gradual accumulation of reactive microglia and reactive astrocytes in HD brain tissue throughout the disease course [318,319,320]. PET scans using ^11^C-(R)-PK11195 and ^11^C-PBR28, two tracers that are high-affinity ligands for the peripheral benzodiazepine receptor (PBR) and markers for glial activation, have revealed extensive glial neuroinflammation in the striatal and cortical regions of HD patients that correlated with disease progression [321,322,323,324,325,326].

Microglia undergo morphological changes and become activated upon exposure to inflammatory stimuli and switch from being a promoter of growth and neurogenesis to a motile phagocytic role [317,327,328]. Signatures of microglia activation including pro-inflammatory interleukin 1B, interleukin 6, interleukin 8, TNFα, and MMP have been detected in the HD striatum, globus pallidus, and cortical areas [122,329,330]. Moreover, it has been suggested that mutant huntingtin protein may activate microglia through NF-κB signalling [331,332,333] and the kyurenine pathway [334,335]. Astrocytes also undergo morphological changes (reactive astrogliosis), and studies have found increased numbers of both neurotoxic A1 [336] and neuroprotective A2 astrocytes in HD [337]. It is proposed that neurotoxic A1 reactive astrocytes may increase synaptic concentrations of proinflammatory cytokines and reactive oxygen species, decrease glutamate uptake, and contribute to membrane depolarisation through impaired potassium homeostasis [336,338,339]. Conversely, neuroprotective A2 type astrocytes produce antioxidants and initiate reconstruction of damaged neuronal circuits [340,341].

#### Treatments Targeting Neuroinflammation

The small molecule compound Neflamapimod was of interest for its ability to inhibit proinflammatory kinases including tumour necrosis factor and interleukin-1β and reduce the number of reactive microglia [342,343]. Neflamapimod has been shown to reduce synaptic dysfunction and slow disease progression in Alzheimer’s disease patients [342,343]. Unfortunately, a phase II clinical trial comparing Neflamapimod against a placebo in 15 HD patients was prematurely terminated due to the COVID-19 pandemic (www.clinicaltrials.gov, identifier NCT03980938, accessed on 17 August 2023). The antibody VX15/2503 (pepinemab) raised against the transmembrane signalling molecule SEMA4D, which plays a role in regulating the activation of reactive microglia and astrocytes [344], was investigated in a placebo-controlled phase II clinical trial involving 256 patients followed over a period of 18 months [344,345]. Results indicated VX15/2503 resulted in a retainment of brain volume compared to the placebo group and showed improvement in cognitive measures, despite not meeting its primary endpoint [344,345]. Laquinimod, a quinoline-3-carboxamide derivate with anti-inflammatory properties, downregulates expression of the pro-apoptotic molecule Bax and reduces caspase-initiated apoptotic pathways [346]. Treatment with Laquinimod reduced striatal and cortical atrophy and improved motor function in an HD mouse model [347,348]. However, a phase II placebo-controlled clinical trial with Laquinimod in 15 patients over a follow up period of 12 months showed no significant change in ^11^C-PBR28 neuroinflammation marker levels between treatment and placebo groups [349]. The antibiotic minocycline was assessed for its anti-apoptotic effects and inhibition of reactive microgliosis [350,351] in a phase III placebo-controlled clinical trial involving 114 patients followed over 18 months (IND 60943). Unfortunately, minocycline did not achieve a >25% improvement in total functional capacity (measure of HD progression) and therefore was not continued in further trials [352].

### 6.5. mHTT Protein Misfolding and Endoplasmic Reticulum Stress

Mutant huntingtin protein remains an important pathogenic centerpiece that links neurodegeneration to abnormal protein degradation, endoplasmic reticulum (ER) stress, disrupted cellular trafficking, initiation of apoptotic pathways, and synaptic dysfunction. Toxic mutant huntingtin oligomer species interfere with ER-associated degradation components (ERAD) resulting in ER stress and the activation of the unfolded protein response [353,354]. The inhibition of ERAD leading to the accumulation of unfolded mutant huntingtin proteins and ER stress was observed in HD mouse models and post-mortem HD patients [355,356,357,358].

Lysosome-mediated macroautophagy plays a key role in the removal of misfolded proteins including mutant huntingtin [359]. Mutant huntingtin has been shown to stimulate the formation of autophagosomes through the inactivation of the mTOR kinase pathway [360,361]. However, despite the increase in autophagosome formation, cargo loading into autophagosomes is impaired in HD [361,362]. Therefore, despite an increase in the number of autophagic vesicles, mutant huntingtin and damaged organelles are not being degraded and instead are left to accumulate in the cytoplasm, where they continue to be toxic [363].

#### Treatments Directly Targeting mHTT Protein

Inhibition of the mTOR pathway induces mutant huntingtin protein autophagy and has demonstrated neuroprotective effects in HD cell and animal models. Rapamycin, an inhibitor of mTOR, reduced mutant huntingtin aggregates and neuronal atrophy in HD fly and mouse models [360]. Rapamycin has also been combined with lithium as a possible treatment option. Lithium induces autophagy through an mTOR-independent mechanism by inhibiting inositol monophosphatase. This dual approach seems to produce additive effects in an HD fly model [364]. Another antagonist to mTOR, metformin, reduced mutant huntingtin protein aggregation and reduced early behavioural defects in an HD mouse model [365]. In HD patients, metformin was associated with an improvement on cognitive tests but did not show a significant improvement in motor function [366].

The neuroprotective sigma-1 receptor is a molecular chaperone that is activated following ER stress [367,368]. Pridopidine, a sigma-1 receptor agonist was shown to increase brain-derived neurotrophic factor (BDNF) levels, reduce mutant huntingtin protein aggregation, and reduce motor impairments in an HD mouse model [369]. Early clinical trials suggested significant improvement to motor function [370,371,372]; however, a placebo-control involving 397 HD patients followed over a period of 26 weeks showed no improvement in UHDRS motor scores and the results of previous trials were not replicated [373].

Other studies have focused on stabilising the mutant huntingtin protein, thereby reducing its toxicity and propensity to form aggregates. Through screening of a library of natural compounds, Wanker and colleagues identified Epigallocatechin-3-gallate (EGCG), a catechin commonly found in green tea, which reduced mutant huntingtin misfolding and aggregation in vitro and decreased neurodegeneration in an HD fly model [374]. A completed phase II placebo-controlled clinical trial involving 54 patients followed over 12 months is expected to publish results (www.clinicaltrials.gov, identifier NCT01357681, accessed on 17 August 2023). Increased transglutaminase activity is thought to contribute to the misfolding of mutant huntingtin and formation of aggregates [375]. Cystamine, a competitive inhibitor of transglutaminase demonstrated improved motor function and overall survival in an HD mouse model, although the quantity and size of aggregates remained unchanged [376]. Unfortunately, a phase II/III placebo-controlled trial investigating cysteamine, the reduced form of Cystamine, in 96 HD patients followed over a period of 18 months showed no statistically significant improvement motor function [377]. Mutant huntingtin aggregation may be due to covalent attractions with metal ions, copper, and iron. Metal chelators including the 8-hydroxyquinoline analog named PBT2 showcased efficacy in an HD mouse model with improvements to motor deficits and extended lifespans [378]. A completed phase II placebo-controlled clinical trial involving 109 patients over a period of 26 weeks established safe and effective dose ranges for further evaluation in larger trials [379].

## 7. Invasive Therapeutics

Deep brain stimulation (DBS) for HD is primarily targeted at the globus pallidus and provides symptomatic relief for chorea. The exact mechanism of DBS is not well understood; however, studies suggest DBS alters electrical and neurochemical activity of neurons near the input electrode, which may confer a local or network-wide effect [380]. An initial pilot study with three individuals showed a reduction in chorea and improvement in quality-of-life scores but no significant effect on dystonia [381]. A phase II clinical trial was recently completed that examined the efficacy of DBS in 48 HD patients (www.clinicaltrials.gov, identifier NCT02535884, accessed on 17 August 2023). Another invasive technique, extracranial stereotactic radioablation of the thalamic region, has shown effectiveness in alleviating tremor symptoms present in Parkinson’s disease [382,383]. A clinical trial is currently evaluating the safety and effectiveness of extracranial stereotactic radioablation of the thalamus or globus pallidus in HD patients (www.clinicaltrials.gov, identifier NCT02252380, accessed on 17 August 2023).

Stem cell transplant therapies aim to replace damaged neurons in affected HD brain areas with new neurons derived from stem cells. Sources of stem cells include mesenchymal stem cells from bone marrow [384] or adipose tissue [385], embryonic stem cells from in vitro fertilized eggs [386], neural stem cells from brain tissue [387], or induced pluripotent stem cells [388]. The safety and efficacy of bone marrow-derived stem cells was evaluated in a phase I/II clinical trial involving 50 participants (www.clinicaltrials.gov, identifier NCT01834053, accessed on 17 August 2023). Results from the trial have not yet been released. An ongoing phase I/II dose escalation trial is investigating the effects of three intravenous injections of stem cells spread across 3 months. Individuals will be monitored for up to five years for safety and efficacy (www.clinicaltrials.gov, identifiers NCT03252535, NCT02728115, accessed on 17 August 2023). Another ongoing clinical trial is evaluating the efficacy of autologous stem cell isolates from adipose tissue in multiple neuropathies including HD (www.clinicaltrials.gov, identifier NCT03297177, accessed on 17 August 2023).

## 8. Genetic Mechanisms and Therapeutics

Huntingtin and mutant huntingtin interact with a wide variety of transcription factors, chromatin remodelling proteins, basal transcriptional machinery, and non-coding RNAs. Both wild-type huntingtin and mutant huntingtin can directly bind to DNA at promoter sequences or intronic/intergenic regions to alter gene expression [148,389]. Transcriptional dysregulation is not restricted to areas of the brain, but rather the ubiquitous expression of *HTT* alters transcript profiles across non-neuronal tissues such as blood [390], muscle [391,392], and liver [53]. Gene expression studies including microarray profiling [393,394], RNA-seq [395], and, recently, single cell RNA-seq [396,397,398] studies have identified several differentially expressed genes between HD and controls in post-mortem brain, HD patient cell lines, and HD animal models. Nevertheless, it is important to note that not all changes in gene expression in HD play a role in pathogenicity and that they may instead reflect the activation of compensatory mechanisms to offset toxic effects [399,400]. The key players of HD transcriptional dysregulation will be discussed in detail, followed by a summary of treatments targeting this mechanism.

### 8.1. BDNF

The brain-derived neurotrophic factor (BDNF) gene encodes a small nerve growth factor (neurotrophin) protein that promotes neuronal growth, maturation, survival, and synaptic activity [401,402]. Loss of BDNF protein is well documented in post-mortem HD brains and can lead to impaired synaptic transmission and loss of neuronal plasticity and contribute to medium spiny neuronal death [114,115]. Depletion of cortical BDNF levels have been shown to exacerbate HD symptoms in mouse models [115]. In contrast, overexpression of BDNF rescued striatal atrophy and ameliorated motor dysfunction in the same mouse models [403]. The loss in BDNF was proposed to be due to a loss of function of wild-type huntingtin protein. Wild-type huntingtin was observed to promote the expression of BDNF and regulate its transport to synapses [401,404]. BDNF and its associated trophic pathways have been key targets for therapeutic development [405].

### 8.2. CREB/CBP

Numerous reports have described transcriptional dysregulation of the CREB (cAMP response element binding protein) pathway in HD [406,407]. CREB belongs to a family of basic leucine zipper transcription factors that play a role in a wide range of physiological processes that include neuronal plasticity, long term memory function, neuronal differentiation, and survival [408]. Additionally, coactivators of CREB, notably CREB-binding protein (CBP) and p300, play important roles with its acetyltransferase activity and ability to recruit and initiate transcription factors such as p53, c-Jun, c-myc, E2F, and the RNA polymerase II transcriptional machinery [148]. In HD, dysregulation of CREB-mediated transcription and/or activity of its cofactors causes a reduction in BDNF levels [409], altered NMDA signalling, and diminished activation of the MAPK/ERK and protein kinase A pathways [408,410]. Importantly CBP has been shown to sequester with mutant huntingtin protein in nuclear inclusions, which may in turn reduce CREB transcriptional activity. This sequestration was hypothesised to increase striatal cell vulnerability in HD mouse models [406,411].

### 8.3. REST

RE1 silencing transcription factor (REST) (also known as neuron-restrictive silencer factor, NRSF) is a Krüppel-type zinc finger protein that acts as a transcriptional repressor of neuronal genes in non-neuronal cells and is important for cell fate and the maintenance of cell identity [412]. In HD, REST has been shown to translocate to the nucleus, where it represses cortical transcripts including BDNF and a variety of HD-related non-coding RNAs (miR-9, miR-124) [401,413]. Wild-type huntingtin protein has been shown to inhibit REST activity by a sequestration mechanism, resulting in the retainment of REST in the cytoplasm. This mechanism is reduced with mutant huntingtin and results in increased translocation of REST into the nucleus and repression of its downstream effectors [413].

### 8.4. PGC1α

Peroxisome proliferator-activated receptor gamma coactivator 1-alpha (PGC-1α) is a transcriptional co-activator that regulates mitochondrial biogenesis, energy homeostasis, and antioxidant defences [414]. Involvement of PGC-1α in HD was first suggested by the observation that PGC-1α knockout mice exhibit mitochondrial dysfunction, striatal neurodegeneration, and motor deficits resembling HD [415,416]. Polymorphisms in PGC-1α have been shown to modify the age of HD onset [417,418], and decreased expression of PGC-1α was observed in post-mortem brains of HD patients and in HD mouse models [419,420]. Further, striatal lentiviral administration of PGC-1α in an HD mouse model ameliorated neurodegeneration and mutant huntingtin aggregation [419]. The decrease in PGC-1α expression was proposed to be due to the effects of mutant huntingtin, which interferes with the CREB/TAF4-mediated activation of PGC-1α by inhibiting the ability of CREB/TAF4 to bind to the PGC-1α promoter.

### 8.5. Epigenetic Modifications

HD epigenetic dysregulation has become increasingly documented, with mutant huntingtin protein implicated in altering the activity of chromatin remodelling enzymes. Changes in chromatin status attributable to histone modifications and DNA methylation/hydroxymethylation have been characterised.

Histone acetylation regulated by histone acetyltransferases (HATs) and histone deacetylases (HDACs) promotes the formation of euchromatin and recruitment of the basal transcriptional machinery. Histone hypoacetylation is widespread in HD, with dysregulation seen in histone acetyltransferases CBP, pCAF, and p300 [148]. Mutant huntingtin protein can bind and potentially sequester HATs, altering the nuclear distribution of HATs and inhibiting their ability to influence chromatin structure [407,411]. Other histone marks including histone H3 lysine 9 (H3K9) are hypermethylated in HD patients, and this has been associated with gene repression [421,422,423]. Conversely, histone H3 lysine 4 (H3K4) marks were hypomethylated in HD patients, and this was attributed to the effects of lysine demethylase 5C (KDM5C). Knockdown of KDM5C alleviated HD-induced toxicity in HD mouse and HD fly models [423].

DNA methylation involves the addition of a methyl group to carbon 5 of cytosine (5-methylcytosine, 5mC) catalysed by DNA methyltransferases [424]. 5mC modifications are largely located in spans of CpG dinucleotides, usually found in gene promoters, and are associated with gene repression [425]. An extra addition of a hydroxyl group onto 5mC via oxidation by ten-eleven translocation enzymes, generating 5-hydroxymethylcytosine (5hmC), is thought to relieve this transcriptional repression [425]. Genome wide investigations in HD mouse striatal cell lines revealed a bias towards 5mC hypomethylation that was enhanced in CpG poor regions. Chromatin immunoprecipitation sequencing (ChIP-seq) showcased the association of transcription factors such as SOX2 and members of the AP-1 gene family with DNA methylation changes [426]. Global levels of 5hmC were also found to be reduced in HD mouse striatum and cortex regions [427]; however, the subset of genes examined in the study was too small to conclude any correlation with HD pathogenesis.

The 21–23 nucleotide long microRNAs (miRNAs) [428,429] are antisense oligonucleotides that bind to mRNA and reduce their abundance [430]. The microRNAs miR-9 and miR-124 were downregulated in the striatum and cerebral cortex of HD patients, causing altered expression of *REST* [431,432,433]. The *HTT* gene itself transcribes an antisense non-coding RNA known as *HTT-AS* that contains three exons, including the repeat tract of exon 1, and is highly expressed in the brain [434]. Overexpression or silencing *of HTT-AS* has been shown to downregulate or upregulate, respectively, the wild-type *HTT* transcript levels [434].

### 8.6. Treatments Targeting Transcriptional Dysregulation

The short-chain fatty acid class of HDAC inhibitors, comprised of sodium butyrate, phenylbutyrate, and valproate, has shown efficacy in HD cell and mouse models [435,436,437]. Clinical trials conducted showcased a mixed response in HD patients, with the trials complicated by limitations including small sample size, different stages of disease, concurrent drug therapies, non-double blinded designs, and short treatment durations [438]. Other HDACs, Suberoylanilide hydroxamic acid (SAHA) and HDACi4b, attenuated motor impairments in HD mouse models [439,440,441,442]; however their therapeutic potential has not been examined in clinical trials. Mithramycin, an FDA approved anti-tumour antibiotic which interferes with the activity of the Sp family of transcription factors (notably Sp1 and Sp3) and inhibits downstream ESET-mediated histone methylation, was able to reduce histone 3 lysine 9 hypermethylation and improved motor and behavioural deficits in an HD mouse model [421,422]. Clinical trials have yet to be performed.

### 8.7. DNA Targeting Approaches

DNA approaches directly target the *HTT* DNA sequence, altering its transcriptional activity or completely removing the deleterious mutation by genomic editing. Zinc finger proteins (ZFP) contain a DNA-binding motif that can be modified to bind to certain regions of interest. ZFP are often fused to transcriptional activators/repressors to alter target gene expression or fused to a nuclease (usually FokI) to cleave at specific sites for precise gene editing [443]. Zinc finger transcriptional repressors targeting the expanded CAG repeat have shown success in patient-derived fibroblasts and stem cell-derived human neurons with up to 90% repression of *mHTT* and less than 15% repression of the wild-type *HTT* [444]. Experiments in HD mouse models also demonstrate efficacy, exhibiting similar levels of repression, and were able to ameliorate several cognitive and motor deficits [444]. Transcription activator-like effectors (TALE) are other DNA-targeting agents comprised of a TAL effector DNA binding domain that can be engineered to bind specific locations. Similarly to ZFP, TALE are often fused to either a transcriptional activator/repressor or nucleases (usually FokI) [445]. TALE fused to a transcriptional repressor designed against *mHTT* allele specific SNPs (rs762855, rs3856973, rs2024115) was able to achieve allele selectivity in human HD fibroblasts [446].

The CRISPR-Cas9 system has been exploited to cleave DNA at a specific location of choice mediated by a synthetic guide RNA (gRNA) apparatus. An *mHTT* allele selective CRISPR-Cas9 approach utilising dual gRNAs specific for *mHTT* SNPs was successful in removal of the promotor region and transcription start site and the polyglutamine expansion of the *HTT* gene and produced permanent inactivation of the *mHTT* allele in patient-derived fibroblasts [447]. Further, CRISPR-Cas9-catalysed selective excision of the *mHTT* allele in Q140 HD knock-in mice attenuated early neuropathology and motor deficits and depleted the formation of aggregates; however, surprisingly, overall survival was not affected [448].

### 8.8. RNA-Targeting Approaches

An alternative approach is to target the *HTT* transcript. RNA interference (RNAi) utilizes the natural defence system against invading pathogenic RNA molecules to ‘silence’ a gene of interest. Double-stranded RNA (including small interfering RNA, siRNA; short hairpin RNA, shRNA; and microRNA, miRNA) complementary to the transcript of interest can be synthesised and introduced into the cell where it binds to the mRNA and targets it for degradation by argonaute 2 [449,450]. In HD, double stranded RNA has been designed to bind the *HTT* transcript (non-selective) or exclusively to the *mHTT* allele via the expanded CAG repeat or regions with allele-specific polymorphisms or small insertions and deletions (selective). Intrastriatal injection of adeno-associated virus (AAV)-containing non-selective *HTT*-targeting siRNA, shRNA, or miRNA into HD mouse models showed efficacy in ameliorating HD phenotypes [451,452,453,454]. A phase I/II proof of concept trial is currently recruiting HD patients to evaluate AAV-mediated delivery of a non-selective *HTT*-targeting miRNA (AMT-130) (www.clinicaltrials.gov, identifier NCT04120493, accessed on 17 August 2023) [455,456].

Antisense oligonucleotides (ASO) are 8–50 nucleotide-long single-stranded DNA molecules synthesised to bind existing pre-mRNA of a targeted gene and mark it for degradation by ribonuclease H [449,457]. The success of ASO in the treatment of genetic diseases has been shown in two FDA approved therapies: Nusinersen for the treatment of spinal muscular atrophy [458] and eteplirsen for the treatment of Duchenne’s muscular dystrophy [459]. In HD, a non-selective ASO termed RG6042 (previously IONIS-HTT_RX_) demonstrated efficacy in three different HD mouse models (YAC128, R6/2, BACHD) with approximately 80% reduction of *mHTT* mRNA and 63% reduction of mutant huntingtin protein. Reversal of the disease phenotype was also documented in all three HD mouse models [460]. These results were further replicated in a large mammalian primate model, which likewise demonstrated 50% reduction of mHTT mRNA in the cortex and 20% reduction in the striatum [460]. Following these promising results, a phase I/II dose escalation trial was carried out with forty-six individuals receiving either intrathecal injections of RG6042 ASO or placebo over a four month period [461]. A 40% reduction in mHTT in the CSF was observed that was dose dependent, possibly reflecting a 55–85% reduction of cortical mutant huntingtin protein and 20–50% reduction of striatal mutant huntingtin protein [461]. A subsequent phase III trial was undertaken with 791 HD individuals placed into 3 groups, with intrathecal injection of RG6042 every 8 weeks, intrathecal injection of RG6042 every 16 weeks, or placebo. Unfortunately, RG6042 was found to exacerbate disease symptoms in the group receiving RG6042 every 8 weeks, with consistently worse scores on UHDRS and total functional capacity compared to the placebo. The individuals given RG6042 every 16 weeks showed no significant differences from the placebo. With the results suggesting that RG6042 was not benefiting patients, dosing was discontinued in March 2021 (www.clinicaltrials.gov, identifier NCT03761849, accessed on 17 August 2023). A dose-finding phase II clinical trial was subsequently initiated to evaluate the safety and efficacy of RG6042 in younger patients with a lower disease burden determined by the CAG-repeat age product score [462].

Allele-specific treatments aim to avoid the complications derived from a reduction in wild-type *HTT*. Two clinical trials have been completed investigating ASO-targeting HD-associated SNP rs362307 (www.clinicaltrials.gov, identifier NCT03225833, accessed on 17 August 2023) and ASO-targeting HD-associated SNP rs362331 (www.clinicaltrials.gov, identifier NCT03225846, accessed on 17 August 2023). However, neither ASO was associated with a significant decrease in mutant huntingtin protein levels in patients’ cerebrospinal fluid compared to the placebo group.

**Table 3 ijms-24-13021-t003:** Huntington’s disease pharmacological therapeutics.

Drug Name	Mechanism of Action	Current Status (Highest Phase)				Clinical Trial Number	Supporting Reference
		I	II	III	IV		
Excitotoxicity							
Amantadine	NMDA inhibitor		?			NCT00001930	[263,264]
Dimebon (laterepirdine)	NMDA inhibitor			X		NCT00497159, NCT00920946, NCT00387270, NCT01085266, NCT00931073	[463]
BN82451	Na+ channel inhibitor		X			NCT02231580	[267]
Memantine	NMDA inhibitor				X	NCT01458470, NCT00652457	[464]
Riluzole	Na+ channel inhibitor			X		NCT00277602	[266]
Dopaminergic pathway							
Bupropion	Dopamine receptor antagonist		X			NCT01914965	[465]
Deutetrabenzine	VMAT2 inhibitor				✓	FDA-approved	[286]
Olanzapine, Tiapride	Dopamine receptor antagonist			/		NCT00632645	
OMS643762	Phosphodiesterase inhibitor		X			NCT02074410	
PF-02545920	Phosphodiesterase inhibitor		X			NCT02342548, NCT02197130, NCT01806896	
Risperidone	Dopamine receptor antagonist		/			NCT04201834, NCT04071639	
Rolipram	Phosphodiesterase inhibitor	✓				NCT01602900	[291]
SOM3355	VMAT2 inhibitor		/			NCT03575676, NCT05475483	
Tetrabenazine	VMAT2 inhibitor				✓	FDA-approved	[284,285,466]
Valbenazine	VMAT2 inhibitor			✓		NCT04102579, NCT04400331	[288]
Mitochondrial dysfunction							
Creatine	Antioxidant			X		NCT04400331, NCT04102579, NCT01412151, NCT01411150, NCT01411163, NCT00712426, NCT00592995	[311,312]
Coenzyme Q10	Cofactor in electron transport chain			X		NCT00608881, NCT00920699	[308]
Eicosapentaenoic acid	Omega-3 fatty acid			X		NCT00146211	[305,306]
Fenofibrate	PPARα agonist		/			NCT03515213	
Resveratrol	Anti-fungal agent			/		NCT02336633	
Triheptanoin	Substrates for Krebs cycle		/			NCT02453061	[315]
Neuroinflammation							
Laquinimod	Reduces expression of Bax		X			NCT02215616	[346,349]
Neflamapimod	p38α MAPK inhibitor		X			NCT03980938	
VX15/2503 (pepinemab)	SEMA4D inhibitor		?			NCT02481674	[345]
Minocycline	Antibiotic			X		NCT00277355, NCT00029874, IND 60943	[352]
mHTT aggregation and autophagy							
Cysteamine	Transglutaminase inhibitor			X		NCT02101957	[377]
Epigallocatechin Gallate	Antioxidant		/			NCT01357681	[374,467]
Metformin	Inhibitor of mTOR			?		NCT04826692	[366]
Nilotinib	Selective Bcr-Abl tyrosine kinase inhibitor	/				NCT03764215	
PBT2	Metal chelator		✓			NCT01590888	[378]
Pridopidine	Sigma1-receptor agonist			X		NCT03019289, NCT02494778, NCT01306929, NCT04556656, NCT02006472, NCT00724048, NCT00665223	[370,371,373]
Selisistat	SirT1 inhibitor		X			NCT01485952, NCT01485965, NCT01521585, NCT01521832	[468,469]
Transcriptional dysregulation							
Lithium + Valproate	HDAC inhibitor		?			NCT00095355	[438]
Phenylbutyrate	HDAC inhibitor		?			NCT00212316	[470]
DNA/RNA targeting approaches							
AMT-130	Nonselective miRNA		/			NCT04120493	
Branaplam	Splicing modifier introducing a pseudoexon		/			NCT05111249	[471]
PTC518	Splicing modifier introducing a premature STOP codon		/			NCT05358717	
Tominersen (IONIS-HTTRX, RG6042)	*mHTT* allele non-selective antisense oligonucleotide			X		NCT03761849, NCT02519036, NCT03342053, NCT03842969	[461,472]
WVE-120101, WVE-120102	*mHTT* allele selective antisense oligonucleotide		X			NCT04617860, NCT03225833, NCT03225846	[473]
Other							
ANX005	Targets C1q of the complement system		/			NCT04514367	
Dextromethorphan/quinidine	Morphinan/class I antiarrhythmic agent			/		NCT03854019	
SRX246	Vasopressin 1a antagonist		/			NCT02507284	[474]
Ursodiol	Bile acid	/				NCT00514774	

✓ Trial showed efficacy; / Trial ongoing or results not reported; ? Mixed efficacy; X Trial did not show efficacy or trial was terminated.

## 9. 150 Years of HD Research

Over the course of 150 years from its first description by George Huntington, significant advances have been achieved in our understanding of Huntington’s disease onset, genetic aetiology, symptomatology, and mechanisms of pathogenesis (a timeline of major HD discovery events is presented in Figure 4). Critical linkage studies in a Venezuelan population and other HD patients led to the eventual discovery of the gene and pathogenic mutation in 1993 [5]. This was subsequently followed by the conception of a genetic test and a rating scale for the assessment of motor symptoms (UHDRS). Establishment of HD mouse models has accelerated discernment of potential pathogenic mechanisms, including the discovery of mutant huntingtin inclusions and description of excitotoxicity and mitochondrial dysfunction. Mouse models have also provided a medium for testing potential therapeutic drugs and led to the first human clinical trials [475]. There are now over 200 HD clinical trials operating worldwide (registered on clinicaltrials.gov). To support therapeutic development, HD registries including HSG PHAROS [91], COHORT [92], TRACK-HD [165,476,477,478], PREDICT-HD [94], REGISTRY [95], and ENROLL-HD [96] are now available. For example, Enroll-HD contains a comprehensive repository of longitudinal clinical data and biospecimens for more than 24,000 participants across 21 countries and 159 clinical sites [479]. Clinical trials have culminated in FDA approval of tetrabenazine in 2008 and later deutetrabenazine in 2017 for the treatment of chorea symptoms. Recent research has been focused on unbiased whole-genome approaches including GWAS, RNA-seq, and, more recently, single cell RNA-seq studies which offer greater resolution for gene expression changes and pathogenic mechanisms at work.

In many ways, HD harbours many of the characteristics of a “textbook” genetic disorder. HD is a disorder with a monogenic, autosomal dominant inheritance, and clearly defined mutation that displays complete penetrance. However, we still do not have a full understanding of its pathogenic manifestation, partially owing to the ubiquitous expression of *HTT* across most tissue types and its role in a variety of molecular pathways. Genetic modifiers complicate the age of onset, and recent GWAS studies point to several important genes involved in DNA repair that contribute to the somatic instability of the CAG pathogenic repeat. Current proposed mechanisms outline an insult to the GABAergic medium spiny neurons of the striatum likely resulting from a combination of glutamate excitotoxicity, mitochondrial dysfunction, and neuroinflammation. A toxic gain of function mechanism by aberrant mHTT or a dominant loss of wild-type HTT function is debated, but the truth may lie in a combination of the two. Evidence for peripheral disease is also accumulating, with reports of involvement of organs including the kidneys, heart, liver, skeletal muscle, and spleen, reflecting the ubiquitous expression and function of *HTT*.

HD remains without a curative intervention. Clinical trials overall have been unsuccessful in translating results from HD models to human disease. These unsatisfactory results may be in part due to the diversity of the HD phenotype, with individual variations in symptom profile and disease stage. Further, a lack of a reliable clinical end point for these studies limits the predictivity and efficacy outlook for these therapies. There is a necessity for better HD animal models that capture more complex features of human disease, better HD biomarkers that predicts responses to treatment, and better clinical study design that accounts for biological variability in the heterogenous study populations. Lastly, a challenge for the development of treatments for late onset neurodegenerative disorders is that by the time the patient is symptomatic there has been considerable cell death, which may promote a further cascade of events independent of the primary cause. Therapeutics in symptomatic patients may therefore need to address the primary cause and secondary outcomes. Treatments that address the primary genetic cause may only be partially effective later in the disease process. It is conceivable that treatment to halt disease progression may need to be delivered prior to symptoms, which will be a challenge under the current clinical trials assessment system.

Current HD therapeutics in use are targeted to alleviate major symptoms. These include VMAT2 inhibitors and antipsychotics for the treatment of chorea and movement-related symptoms, NMDA receptor antagonists for the treatment of dementia-related symptoms, antioxidants against mitochondrial metabolic dysfunction, and autophagy inducers to remove toxic mutant huntingtin protein. Recent attention has been focused on genetic intervention. Promising results have been shown by ASOs and RNAis in HD animal models but have not yet been able to translate into human clinical trials. The latest studies have focused on utilizing the CRISPR-cas9 system to excise the pathogenic mutation altogether, which may provide not only symptomatic relief but offer a long-term solution.

## Figures and Tables

**Figure 1 ijms-24-13021-f001:**
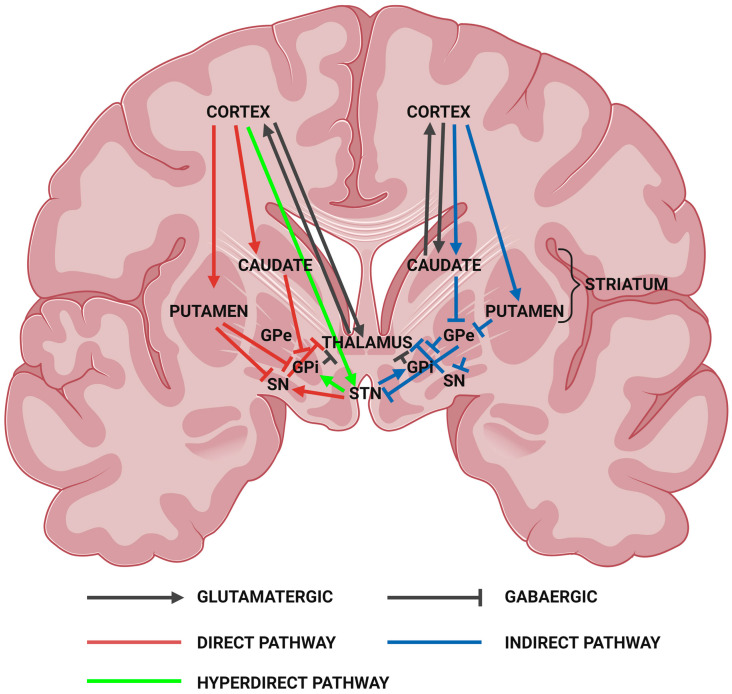
Coronal section of the human brain outlining the basal ganglia. Major signalling pathways of the basal ganglia include the direct, indirect, and hyperdirect pathways. The direct and hyperdirect pathways coordinate excitation of the cerebral cortex. The indirect pathway inhibits cortical excitation. In HD, the medium spiny neurons of the striatum projecting into the globus pallidus are degenerated. GPe, globus pallidus externa; GPi, globus pallidus interna; SN, substantia nigra; STN, subthalamic nucleus. Created with BioRender.com (accessed on 14 January 2023).

**Figure 2 ijms-24-13021-f002:**
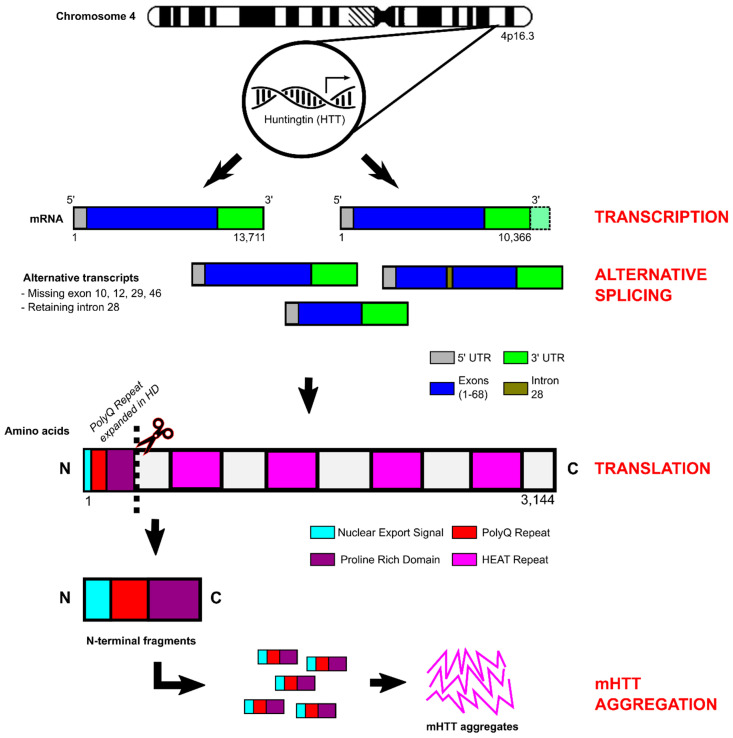
Huntingtin structure and processing. The huntingtin gene (*HTT*) contains 67 exons that are processed to form two mRNA transcripts of 10,366 bp and 13,711 bp that differ in an additional 3′UTR sequence of 3360 bp. Alternative splicing generates several *HTT* isoforms and transcripts lacking exons 10, 12, 29, and 46 or retaining a 57 bp portion of intron 28. Huntingtin protein is comprised of an N-terminal region that encompasses a 17 amino acid huntingtin nuclear export signal, the polyglutamine repeat (polyQ), and a proline rich domain. The remainder of the protein is organised into clusters of anti-parallel alpha-helical HEAT repeats. Proteolytic cleavage of N-terminal fragments by endoproteases, caspases and calpains occurs more frequently in HD, and it has been proposed that these N-terminal fragments are involved in disease pathogenesis.

**Figure 3 ijms-24-13021-f003:**
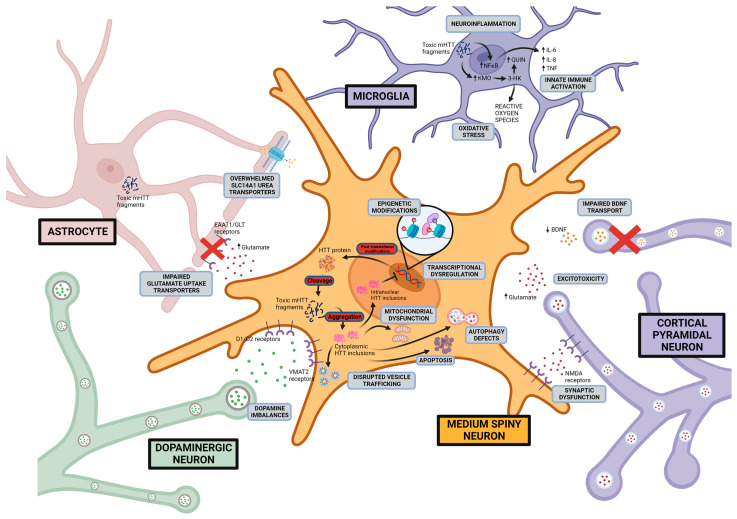
Proposed HD pathogenic mechanisms This figure shows possible Huntington disease mechanisms and the cellular location of the aberrant activity including excitotoxicity, dopamine imbalance, mitochondrial dysfunction, disruption of proteostasis, initiation of apoptotic pathways, transcriptional dysregulation, and neuroinflammation. Created with BioRender.com (accessed on 17 August 2023).

**Figure 4 ijms-24-13021-f004:**
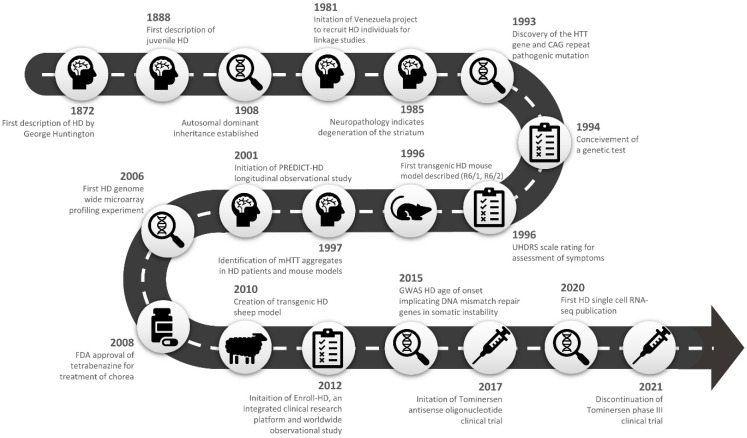
150 years of Huntington’s disease research showcasing research milestones.

**Table 1 ijms-24-13021-t001:** Epidemiology of Huntington’s Disease.

Country	Region	Prevalence (per 100,000)	Year Studied	Reference
Europe				
Russia	Volgograd and Volzhsky	0.6	2000s	[20]
Iceland	-	1.0	2007	[21]
Finland	-	2.12	2010	[22]
Greece	-	3.95	2008	[23]
Croatia	Rijeka	4.5	1981	[24]
Germany	Franconia	4.7	1987	[25]
France	Haute Vienne	4.8	1970s	[26]
	Nord and Pas-de-Calais	5.0	1980s	[27]
Slovenia	-	5.2	2006	[28]
Sweden	-	4.5	1974	[29]
	-	5.6	1985	[30]
Netherlands	-	6.5	1999	[31]
Norway	-	5.8	1950	[32]
	-	6.7	1940	[32]
	-	6.9	1930	[32]
Spain	Valencia	5.38	1992	[33]
	Salamanca	8.4	1980s	[34]
Italy	Molise	10.85	2010s	[35]
Malta	-	11.8	1994	[36]
United Kingdom	Northern Ireland	6.4	1991	[37]
	Northern Ireland	10.4	2001	[16]
	-	11.2	1990	[7]
	-	12.3	2010	[7]
Americas				
Venezuela	Nationwide excluding Zulia	0.5	2006	[38]
Mexico	Mexico City	4.0	2008	[39]
United States of America	Michigan	5.0	1940	[40]
	South Carolina	5.0	1980	[41]
	Maryland	5.15	1980s	[42]
	Minnesota	6.3	1989	[43]
Canada	Quebec	3.4	1963	[44]
	Saskatchewan and Manitoba	8.4	1975	[45]
	British Columbia	13.7	2012	[14]
Oceania				
Australia	New South Wales	6.3	1996	[46]
	Tasmania	12.1	1990	[19]
Africa				
South Africa	-	0.01	1970s	[12]
Zimbabwe	-	0.5–1	1980s	[47]
	Kilimanjaro	7	1970s	[48]
Asia				
China	Hong Kong	0.37	1991	[10,11]
	Taiwan	0.42	2007	[49]
Japan	San-in	0.65	1997	[9]
	Western Japan	0.72	1997	[9]

## Data Availability

Not applicable.

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
