# Peer review of "From Pathogenesis to Therapeutics: A Review of 150 Years of Huntington’s Disease Research"

_ijms, 2023, doi:10.3390/ijms241613021_

Round 1
Author Response
Dear reviewer,
The authors like to thank the reviewer for providing a thorough review and engaging comments. Please refer to the attachment for a point-point response to the comments.
Kind regards,
Andrew Jiang

Reviewer 2 Report
This is a comprehensive and well-written review about Huntington’s disease. Some comments for improvement:
1) Although many aspects are covered, some important ones are missing. For example, there should be a section on oxidative stress and ER stress in HD and therapeutic approaches for this. Also in therapeutic approaches, stem cell based approaches are missing.
2) There should be a discussion on what is known about why the brain and especially the striatum are affected and not or much less other organs, while mHtt is ubiquitously expressed.
3) The style is quite flat, it should be improved to make the review more interesting. For example in section 4.2, the size of the Htt protein can be qualified as quite large. “Huntingtin is ubiquitously expressed throughout most human tissues but is highest in the nervous system…”, this should be mentioned as quite surprising, given the effects only in the CNS. etc.
4) Abstract: “…highly penetrant dominant disorder acting through either a loss of wild-type HTT function or a toxic gain of function…”. I don’t think the loss of function and gain of toxic function should be put in an equal standing in the abstract. Though loss of function contributes, gain of toxic function is shown to be responsible for the main pathogenicity.
5) Table 1 should be sorted by descending prevalence.
6) Page 5 “…helical structure known as HTTNT important for HTT nuclear export and retention in the endoplasmic reticulum…”. HTT is not an ER or secretory protein, it is not retained in the ER,
7) Page 8 “Conversely, aggregation may not have such a pernicious effect…”. The formation of oligomers can still be the cause and are not detected through common microscopy.
8) Page 10 “HD is a late onset disorder with an approximate 40-year presymptomatic period. The short lifespan of mice…” It should be mentioned that this is in humans, in animals it might be much shorter.
9) The fonts in figures 3 and 4 are too small.
10) Page 15 “Pridopidine, a dopamine stabiliser was reported to increase…”. The information about pridopidine should be updated with the newer literature, which should be cited. It is considered now a Sigma-1 receptor agonist, not a dopamine stabiliser. The new clinical trial with this drug should also be mentioned.
The English is very good. I spotted some typos such as “reminder” in page 6, which should be “remainder”. The text should be rechecked carefully.
Author Response

(The authors gave the same response as above.)

Reviewer 3 Report
This review article summarizes the pathophysiological mechanisms of Huntington’s disease (HD), including mitochondrial dysfunction, dopamine abnormalities, inflammatory response, transcriptional dysregulation, and excitotoxicity. Moreover, the authors have listed a number of clinical trials and they have discussed different potential therapeutic strategies for the treatment of HD. Overall, the review is interesting in its field, is well presented and discussed, and provides useful information. However, in my opinion, the paper needs rewriting and reworking to make it appealing to potential readers. In addition, the authors should address some minor compulsory issues:
This is a complete review article and it has a wide enough scope for readership. But to make the topic of the study more focused on HD, I suggest reorganizing the sections.
Some sections can be easily mixed, which will make the review more consistent and easier to read.
There are two “Table 2”. “HD pharmacological therapeutics” should correspond to Table 3.
The toxicity of diverse forms of HTT should be discussed.
The authors need to reduce the number of references.
Section 18.3 (Treatments directly targeting mHTT protein) does not belong to “Genetic therapeutics”.
The substantia nigra is missing in Fig.2.
Spell out the names before using the abbreviations (e.g., MSNs).
Author Response

(The authors gave the same response as above.)

Round 2
Reviewer 3 Report
The authors have satisfactorily addressed all my concerns.